# Compartmentalized Quantitative Analysis of Concrete Sulfate-Damaged Area Based on Ultrasonic Velocity

**DOI:** 10.3390/ma16072658

**Published:** 2023-03-27

**Authors:** Yinghua Jian, Dunwen Liu, Kunpeng Cao, Yu Tang

**Affiliations:** School of Resources and Safety Engineering, Central South University, Changsha 410083, China

**Keywords:** sulfate erosion, concrete, ultrasonic speed measurement, damage assessment, CT scanning

## Abstract

The corrosion of concrete in sulfate environments is a difficult problem in the durability of civil engineering structures. To investigate the variability of deterioration damage to concrete structures by sulfate erosion under non-destructive testing and quantify the protective effect of silane coatings on concrete under the action of sulfate erosion, an accelerated erosion experiment was carried out using field sampling in a tunnel project under a sulfate erosion environment. By means of ultrasonic velocity measurement and CT scanning, the samples protected by a silane coating under the sulfate attack environment were compared with those not protected. The deterioration characteristics of concrete under the sulfate attack environment and the protective effect of silane coating on the concrete structure were analyzed. In addition, a method for evaluating the sulfate damage to concrete based on CT images and ultrasonic velocity analysis was proposed. The results show that the samples prepared in the field show a significant difference in ultrasonic velocity in the process of erosion and deterioration according to the material difference at the measuring point interface. Through the overall damage evaluation analysis of the sample, it is concluded that the damage degree of the protected group sample is light and the heterogeneity is weak, whereas the local damage to the exposed group is serious. Combined with the CT image analysis of concrete before and after loading, the distribution characteristics of the damaged area divided by the concrete sulfate damage evaluation method proposed in this paper are highly similar to the real situation. The results of the study can provide a reference for similar projects for the detection, analysis, protection and evaluation of sulfate-attacked concrete.

## 1. Introduction

In recent years, the influence of concrete diseases caused by sulfate on the durability of concrete structures has attracted more and more attention. In general, sulfate penetrates into the interior of the concrete with water as the medium and causes structural deterioration through mechanical or chemical interaction with the cement slurry [1,2], thus reducing or losing the carrying capacity. The engineering environment, material properties, erosion concentration and other factors will all have an impact on the erosion process. Even in practical engineering, the same structure has multiple types of erosion [3,4,5], which makes it difficult to infer the deterioration of concrete performance from the perspective of the mechanism. Due to its convenience and non-destructive properties, structural and acoustic testing techniques have become important tools for researchers to reveal and analyze the performance of concrete deteriorated by sulfate attack [6,7,8].

As a non-destructive testing technology, ultrasonic velocity measurement can be used to evaluate the performance of concrete [9,10,11]. When the internal structure of concrete changes, its ultrasonic velocity will also change accordingly, and the change in ultrasonic velocity of the sample is usually regarded as an important index to measure the integrity of the sample [12,13]. There are many application scenarios of ultrasonic testing in concrete performance analysis. For example, C-S-H gel formed by cement hydration can lead to the continuous compaction of the concrete structure and decrease of concrete porosity by connecting internal aggregate and hydration products. Ultrasonic velocity can be used to measure hydration velocity and material state [14,15]. Once the internal damage to concrete worsens, the cohesion of concrete decreases, and its ultrasonic velocity will also show a downward trend [16]. In addition to the direct use of ultrasonic velocity as an evaluation index, the intermediate variable based on ultrasonic velocity is also an effective index to evaluate concrete performance. For example, Ikumi [17] used the envelope area of the curve of ultrasonic velocity variation as a way to measure the sample damage in concrete sulfate damage. Chu [18] constructed the ultrasonic velocity attenuation coefficient in a viscoelastic medium to represent the damage state of concrete, and the results showed that the magnitude of the concrete attenuation coefficient was positively correlated with the damage degree of samples. Carrion [19] built a damage evaluation model for concrete materials based on ultrasonic velocity and dynamic elastic modulus combined with the recursive graph quantitative analysis method. However, if only the ultrasonic velocity measurement method is used to analyze the performance of concrete, the result is often rough because of the strong heterogeneous influence of concrete. 

As the structural change of concrete after damage has an important impact on its performance, the testing and analysis technology of concrete structure becomes a powerful support in the research of concrete damage. In various structural tests, CT scanning is extremely suitable for studying the evolution law of the internal microstructure of concrete samples in the process of erosion because it can continuously record the internal images of the samples [20,21,22]. With the help of digital model reconstruction technology, CT scan images can be restored into visual data models, which have positive significance for the study of the erosion process of the whole sample. For example, Yang [23,24] used X-CT technology to study the damage evolution of the internal structure of mortar samples under full-soaking and half-soaking conditions, and Zhang [25] studied the characterization and analysis of the change of concrete structure under uniaxial loading based on the fractal dimension of CT images. Chen [26] studied the concrete damage process under the combined action of chlorine salt erosion and the freeze–thaw cycle. This kind of research can directly reflect the internal structure state of the sample and reveal the evolution and development process of defects, but it is difficult to directly quantify the performance changes in concrete damage only based on image data. Combined with CT scanning image and ultrasonic velocity measurements, a CT image is used to extract the technical advantages of its internal junction characteristics [27], and more accurate ultrasonic velocity change data can be obtained by eliminating stable components, such as coarse aggregate [28], to realize the refined analysis of ultrasonic velocity on the concrete damage process [29].

Non-destructive testing techniques, such as ultrasonic velocity measurement and X-CT, have played an important role in analyzing the properties of cement mortar and concrete subjected to sulfate attack under indoor conditions. It is worth noting that diseases in real engineering are often more complex and inhomogeneous than in laboratory experiments [28,29]. The inhomogeneity of this disease was also observed in sulfate attack experiments on concrete samples extracted from service structures [30,31]. The author believes that the ultrasonic velocity test can reflect the regional difference of concrete damage caused by sulfate attack to realize the evaluation of concrete deterioration by ultrasonic velocity damage. At the same time, combined with CT image analysis, the evaluation results can be tested to a certain extent. This work can be used as a reference for testing and evaluating concrete properties in sulfate environments.

## 2. Materials and Methods

### 2.1. Sample Preparation and Experimental Design

The sample selected in this experiment is from an in-service cross-river tunnel segment, whose designed strength grade is C50, 28-day compressive strength is 61.6 MPa, and water–cement ratio is 0.4. The concrete is made of P.O 42.5 grade ordinary Portland cement with 28-day flexural strength of 7.5 MPa and compressive strength of 46.8 MPa. Fly ash for use in concrete application specification GB/T 1956-2017 was Class F-II fly ash, with fineness of 45 μm sieve and residue 15.2%. The specific mix ratio parameters are shown in Table 1.

The steel bar was extracted after the segment was broken, and the cylinder sample of 50 mm × 100 mm was extracted and cut using the core drilling equipment. At the beginning of the test, the sample age was 1150 d, and the original compressive strength was 75 MPa.

Before the original tunnel segments were put into use, isooctyltrioxyethylsilane was used to spray protective coating on the surface of the segments, and the designed dosage on site was 450 g/m^2^.

In this paper, the samples were divided into protected group, exposed group and control group. Before the experiment, the protection group was coated with silane coating according to the designed dosage, and the amount of brushing for a single sample was 8.9 g. The surfaces of the exposed group and the control group were not treated.

The curing cycle of dry and wet cycle experiment consisted of a soaking stage (wet stage) and a drying stage (dry stage), and the curing starts from the soaking stage. The erosion process of tunnel segments was simulated by soaking the test blocks in the erosion solution of corresponding components during the experimental soaking period. At the soaking stage, the test blocks of the protection group and the exposure group were placed upright in a PVC box with a lid containing 10% sodium sulfate solution, and the control group was placed in a PVC box also containing water. The distance between the top of the test block and the liquid level in the box was no less than 3 cm, and the solution in the box was replaced every 30 cycles. PVC box was placed in constant temperature and humidity curing box, with setting curing temperature of 20 ± 0.5 °C and relative humidity of 95 ± 0.5%. After soaking the samples for the specified time, they entered the drying stage. The samples were removed from the erosive liquid tank, the residual liquid on the surface was wiped off, and then they were placed in the blast drying oven. After being dried at 60 °C for 6 h, the samples were taken out and cooled in a cool and ventilated place for 2 h and then put back into the original PVC box for the next wetting and drying cycle. Three groups of samples began a cycle at the same time, with each group of samples undergoing cycles 180 times. The sample preparation and curing process are shown in Figure 1. The specific conservation environmental parameters of each group are shown in Table 2.

### 2.2. Sample Test Scheme

In this paper, three methods were integrated to analyze the performance changes of concrete samples in the process of rapid cyclic dry–wet erosion. The whole state of concrete was analyzed using ultrasonic testing method, and the structural changes of the samples were analyzed using CT scanning, and the strength changes of the samples under different conditions were compared using strength test. The overall test scheme of the test is shown in Figure 2.

### 2.3. Ultrasonic Velocity Test and Data Calculation

Starting with the dry and wet cyclic erosion experiment, ultrasonic velocity was tested once every 20 times after dry and wet cyclic curing. Before the test, it was necessary to arrange measuring points for the sample. Based on the base position of the sample, 4 acoustic measuring surfaces H_1_, H_2_, H_3_ and H_4_ were arranged at equal intervals of 2 cm upward. A position on the H_1_ interview sample was selected as the starting point M_11_ of H_1_ acoustic surface measurement, and the measurement points M_1_^2^, M_1_^3^, …, M_1_^18^ were arranged in the side circumferential direction with equal intervals of 20° in a clockwise direction.

The measuring line with M_1_^1^ as the measuring point was denoted as L_1_, and so on; nine measuring lines were arranged in a clockwise direction. The starting points M_2_^1^, M_3_^1^ and M_4_^1^ of other acoustic measuring surfaces were marked at the same vertical position as M_1_^1^, and the remaining measuring points H_2_, H_3_, and H_4_ were arranged according to the same rules as H_1_ and marked with oil pen. The arrangement of measuring points and the testing process are shown in Figure 3.

After marking the test points of the samples, the initial sound velocity was tested one by one for each group of samples. During the test, H_1_, H_2_, H_3_ and H_4_ acoustic surfaces were used for sound velocity test layer by layer. Before testing the ultrasonic velocity of the measuring point, apply an appropriate amount of coupling agent on the surface of the measuring point of the sample and the surface of the measuring probe, and align the center of the transducer with the measuring point. Test from M_i_^1^, and measure the waveform of sound wave passing through the sample in a clockwise direction. The transmitting transducer moves from M_i_^1^ to M_i_^9^ one by one in a clockwise direction, while the receiving transducer moves from M_i_^10^ to M_i_^18^ one by one in a clockwise direction. Each time the two transducers move, a test is conducted accordingly, and the test velocity is calculated. After the test is completed, the residual coupler on the surface should be cleaned with a wet towel, and the samples should be cleaned with the corresponding soaking solution of each group to restore the curing environment on the surface of the samples.

In this paper, the sound velocity test of test measuring line j on measuring surface i for the n times is denoted as $\;V_{i,n}^{j}$.

Then, the average sound velocity of surface i in n  times test can be calculated according to Formula (1):(1)$$\bar{V_{i,n}}=\frac{1}{9}\overset{9}{\underset{j=1}{\sum }}V_{i,n}^{j}$$

The average sound velocity of the first test sample $\bar{V_{n}\;}$ can be calculated according to Formula (2):(2)$$\bar{V_{n}}=\frac{1}{4}\overset{4}{\underset{i=1}{\sum }}\bar{V_{i,n}}$$

In order to analyze the change of velocity of the sample in the wet and dry cycle, the average ultrasonic velocity change rate $\bar{r_{n}}$ of the sample is defined according to Formula (3):(3)$$\bar{r_{n}}=\frac{\bar{V_{0}}-\bar{V_{n}}}{\bar{V_{0}}}\times 100\%$$

In order to analyze the fluctuation of ultrasonic velocity at each measuring point in the dry–wet cyclic erosion experiment, the change rate of ultrasonic velocity $r_{i,n}^{j}$ at a single point is defined to be calculated according to Formula (4):(4)$$r_{i,n}^{j}=\frac{V_{i,0}^{j}-V_{i,n}^{j}}{V_{i,0}^{j}}\times 100\%$$
where $r_{i,n}^{j}$ represents the change of the ultrasonic velocity relative to the original ultrasonic velocity of the sample located on the measuring line j of the measuring surface  i after the first test.

In general, the dynamic elastic modulus $E_{d}$ of materials can be calculated using Formula (5), where v is the longitudinal ultrasonic velocity of materials, μ  is the Poisson’s ratio of materials, and ρ is the density of materials:(5)$$E_{d}=\frac{\left(1+\mu \right)\left(1-2\mu \right)\rho v^{2}}{1-\mu }\times 100\%$$

If the coefficient k is:(6)$$k=\frac{\left(1+\mu \right)\left(1-2\mu \right)\rho }{1-\mu }$$
then Formula (5) is:(7)$$E_{d}=kv^{2}$$

According to Formula (7), the dynamic elastic modulus of materials can be changed by the longitudinal ultrasonic velocity of materials and specific material parameters.

When analyzing the overall dynamic elastic modulus of the sample, the average velocity $\bar{V_{n}}$ is used in this paper, so Formula (7) is:(8)$$E_{d_{n}}=k{\bar{V_{n}}}^{2}$$

Namely, $E_{d_{n}}$ is the dynamic elastic modulus of the sample under n times of loading.

In the process of erosion, due to the constant change of $\bar{V_{n}}$ of the sample, its $E_{d_{n}}$ changes accordingly. For comparative analysis of its damage process, the damage metric of the dynamic elastic modulus $d_{E}$ is defined as:(9)$$d_{E}=\frac{E_{d_{0}}-E_{d_{n}}}{E_{d_{0}}}$$
where $E_{d_{0}}$ is the dynamic elastic modulus of the sample before the dry and wet cyclic erosion, i.e., the initial dynamic elastic modulus.

As can be seen from Formulas (8) and (9), although $d_{E}$ is a measure of the change of dynamic elastic modulus in the process of cyclic dry–wet erosion, its result is not affected by the material characteristics of the sample. When the sample is filled in the early stage of erosion, the dynamic elastic modulus $E_{d_{n}}$ increases, and $d_{E}$ is negative. When the sample deteriorates due to subsequent damage, the dynamic elastic modulus $E_{d_{n}}$ decreases, and $d_{E}$ is positive.

## 3. Results

### 3.1. Strength

The uniaxial compressive strength of each group is shown in Table 3. It can be seen from the table that the strength of the control group is basically consistent with that of the original sample, while the uniaxial compressive strength of both the exposed group and the protection group decreases significantly. Compared to the control group, the compressive strength was reduced by about 27% in the exposed group and 17% in the protective group. The results show that the silane coating can effectively reduce the sulfate damage to concrete but cannot completely avoid the sulfate damage to concrete strength.

### 3.2. Ultrasonic Velocity Variation Characteristics 

According to the calculation methods of Formulas (2) and (3), the average ultrasonic velocity of the three groups of samples in the wet and dry cycle is calculated, and its change and change rate curve is drawn, as shown in Figure 4.

It can be seen from Figure 4a that the average ultrasonic velocity variation trend of the three groups of samples presents certain differences. The exposed group presents a relatively obvious trend of first increase and then decrease. The average ultrasonic velocity of samples from protected group also presents a similar trend, but the change is not as significant as that of exposed group. The average ultrasonic velocity of control group fluctuates back and forth, but the change is small.

Samples of exposed group and protected group both show a slow rising trend at the early stage of the wet and dry cycle. When the cycle is 100 times, the average ultrasonic velocity of the three groups reaches the peak and then begins to decline, while the ultrasonic velocity of the protected group declines gently. However, the final average ultrasonic velocity is still slightly lower than the initial ultrasonic velocity, indicating that the dry and wet cycle experiment has caused some damage to it. In the descending stage of ultrasonic velocity, the exposed group has the fastest velocity change and the lowest velocity, indicating a relatively high degree of deterioration.

As shown in Figure 4b, the average velocity fluctuation of the three groups of samples in descending order is control group, protected group and exposed group. The ultrasonic velocity of the sample in the control group is relatively stable, and its maximum ultrasonic velocity changes less than 2% during the whole dry–wet cycle experiment. The speed change of the exposed group is more significant. In the range from 0 cycles to 100 cycles, the ultrasonic velocity of the exposed group increases, but the increase rate is not obvious, only slightly greater than that of the control group. During the whole experiment, the average velocity increase of the exposed group samples before 100 cycles is the highest in the three groups, while the ultrasonic velocity decreases after 100 cycles, but does not exceed 2%. The increase of ultrasonic velocity in the early erosion stage of the protected group is mainly due to the better surface conditions of the samples in the early curing stage after the application of silane coating, and the erosion components only enter the concrete through some pores. In the early stage, the crystallization of sodium sulfate crystals instead filled the sample to a certain extent.

The sample in this paper is directly made from formed concrete segments, whose structure is not as uniform as that of precast cement mortar or concrete samples in the laboratory, and there are certain differences in the ultrasonic velocity test results of different measuring points during the dry–wet cycle. In this paper, ultrasonic velocity variation curves of representative measuring points in the wet and dry cycles of exposed group, protected group and control group are, respectively, drawn, as shown in Figure 5.

According to the calculation method of Formula (4), the corresponding rate of change of ultrasonic velocity was calculated for each measuring point selected in Figure 5 and the change curve was drawn, as shown in Figure 6.

It can be easily seen from Figure 6a that the final ultrasonic velocity loss of measuring point 35 exceeds 15%, while that of measuring point 7 is only about 1/3 of that of measuring point 35. This indicates that in the dry and wet cyclic erosion experiment of the exposed group, at the point position at the interface between aggregate and cement paste, the physical salt attack becomes the dominant effect on the structure.

As can be seen from Figure 6b, the ultrasonic velocity changes of the three types of feature points in the protected group are smaller than those of the same type in the exposed group. The addition of the silane protective coating better protects the surface of the concrete, thus inhibiting the decline of the ultrasonic velocity of the samples in the dry and wet cyclic erosion experiment. However, compared with the samples without erosion in Figure 6c, the ultrasonic velocity still changed to a certain extent, reflecting the slow decline of the protective effect of silane coating under harsh experimental conditions. 

According to Formula (9), the dynamic elastic modulus damage metric $d_{e}$ of the three groups of samples in the process of cyclic dry–wet erosion was calculated, and its change process is shown in Figure 7.

As can be seen from Figure 7, the variation amplitude of the dynamic mold damage to samples is from small to large: the control group, protected group, and exposed group. The dynamic elastic modulus damage to the control group samples fluctuated back and forth during the whole wet and dry cycle experiment, but the change was weak. The dynamic elastic modulus of the exposed group changes significantly. From 0 to 100 cycles, the dynamic elastic modulus damage to the exposed group decreases all the time, indicating that the dynamic elastic modulus of the sample becomes larger and larger compared to the initial state. After more than 100 dry and wet cycles, the dynamic elastic modulus loss of the sample began to increase rapidly. The dynamic elastic modulus damage to specimens in the protected group decreased to the highest level in the three groups before 100 cycles, while the dynamic elastic modulus began to increase after 100 cycles. Table 4 lists the final dynamic elastic modulus loss of samples in each group at the end of 180 dry and wet cycles. The dynamic elastic modulus damage to the control group is only 0.0075, which can be considered to be roughly equivalent to the dynamic elastic modulus of samples in the original state. The dynamic elastic modulus loss of the exposed group is 0.1196, which is more than 10%, and that of the protected group is only 0.0313, indicating that the silane protective coating provides good protection to the specimen surface.

### 3.3. Methods of Evaluation and Analysis of Damage

As a kind of artificial composite material, the structural heterogeneity of concrete has become an important factor affecting the mechanical properties of the material. However, the random and complex distribution and shape of the crack structure of concrete make the quantitative characterization of the heterogeneous characteristics of concrete become a difficult problem. It can be seen from the previous article that the ultrasonic velocity variation of the sample in each measuring line is different. This difference is due to the slowing down of ultrasonic waves generated by the transducer as they pass through the uneven structure of the concrete. Therefore, the attenuation of measuring line ultrasonic velocity can be used indirectly as a quantitative characterization of concrete structural differences.

In the ultrasonic testing scheme used in this paper, the sample is divided into four thick measuring surfaces, and measuring lines are arranged at equal intervals on each side according to the same rules. At the same time, measuring lines are used as the test basis of the scheme, and each measuring line has the test ultrasonic velocity as the data support, which is taken as the basic unit of the sample region division in this section.

For the unit region divided by the measuring line L_1_ of the measuring surface H_1_ as the axis, the test ultrasonic velocity is taken as the basic data of the analysis. In order to reflect the change of ultrasonic velocity in the wet and dry cycle and eliminate the influence brought by the difference of material composition as far as possible, the dynamic elastic modulus damage $d_{i,j}$ of a measuring line is defined by referring to Formula (9):(10)$$d_{i.j}=\frac{V_{i,0}^{j}-V_{i,n}^{j}}{V_{i,0}^{j}}$$

$E_{i,n}^{j}$ is the dynamic elastic modulus calculated by measuring line j in measuring surface i during the nth test. It can be seen from Formula (8) that $E_{i,n}^{j}$ can be calculated according to Formula (11):(11)$$E_{i,n}^{j}=k{\left(V_{i,n}^{j}\right)}^{2}$$

In this paper, an equal interval arrangement is adopted when setting measuring points. Each test point just crosses the center of the measuring surface circle to form a measuring line, and the included angle between adjacent measuring lines is 20°. In order to divide the whole surface evenly, the radiating range of 10° to both sides is taken as the specific range of influence by taking the measuring line as the center and forming a symmetrical double sector area. According to the same division method as the L_1_ influence area, the influence area of the other eight measuring lines can be drawn similarly, and the equal area of the test plane can be divided into nine parts.

In the process of cyclic dry–wet erosion, the ultrasonic velocity changes of each measuring line are different, and finally, the values of dynamic elastic modulus damage $d_{i.j}$ of the measuring line are different. To measure the distribution characteristics of the dynamic elastic modulus of each measuring line on the same measuring surface, the set $\left\{d_{i,j}\right\}$ of the dynamic elastic modulus defined on the measuring surface i is here, and its range $R_{d}$ is calculated according to Formula (12):(12)$$R_{d}={\left\{d_{i,j}\right\}}_{max}-{\left\{d_{i,j}\right\}}_{min}$$

According to the value of $R_{d}$, the distribution of the dynamic elastic modulus damage to the measuring line can be judged. The range of dynamic elastic modulus damage to each measuring line on the measuring surface can be divided in equal proportion. The distribution of damage to each measuring line can be judged according to the range of each measuring line. In this paper, the measuring surfaces are divided into a uniform measuring surface, dichotomous measuring surface and tripartite measuring surface according to different values of $R_{d}$. Table 5 shows the dividing standard of measuring surfaces.

As the measuring points with the same measuring line number are distributed in the same vertical position among different measuring surfaces, the partition graphs of the four measuring surfaces are superimposed, and the measuring lines with the same measuring line number just coincide. In order to fully refer to the damage analysis results of each plane and then form an intuitive expression of the overall damage distribution of the sample, this paper uses the stacking method of scoring to determine the sample uniformity partition method. For different region types, this paper provides scoring assignments, as shown in Table 6. According to the score and assignment of different regions, after the regional division results of the four groups of measuring surfaces are superimposed, there will be four scores and assignments of the same measuring line area. 

The score and assignment of the sample were obtained according to the measuring line premises. According to the value, the final damaged area of the sample was divided according to the score and assignment division in Table 7. The overall analysis process is shown in Figure 8.

### 3.4. Damage Analysis Results of Samples

The dynamic elastic modulus damage values of all measuring points of the exposed group and the protected group at the end of the wet and dry cycle are calculated according to Formula (10). The range of dynamic elastic modulus of each measuring surface is calculated according to Formula (12), and the classification types of measuring surfaces are determined according to Table 5. The results are shown in Table 8.

Based on Table 7 and Table 8, the value ranges of the dynamic elastic modulus damage to measuring lines in the corresponding areas of each surface of the exposed group and protected group are determined, as shown in Table 9.

The dynamic elastic modulus damage value of each measuring line is obtained in Table 9 to determine the damage zone where each measured line is located, and the damage zone distribution on each measuring surface of samples from the exposed group and the protected group is obtained according to the type of damage zone. The results are shown in Figure 9.

As can be seen from Figure 9, H_1_, H_3_ and H_4_ of the samples from the exposed group are more seriously damaged, while H_2_ is relatively less damaged due to its complete structure and a large amount of aggregate. The exposed group samples also use measuring surface H_2_ as the interface. In the two main damaged areas of measuring surface H_1_, L_4_–L_5_ and L_8_–L_9_, the damage to L_4_–L_5_ extends to measuring surface H_2_, while the damage to L_8_–L_9_ is still mainly concentrated near measuring surface H_1_. The damage to H_3_ and H_4_ is serious. L_6_–L_1_ is the most serious damaged area of the two measuring surfaces, with obvious damage overlapping on both sides, and L_8_ in the core is the most serious. In the exposed group samples, severe damage occurs in all three measuring surfaces within measuring line L_8_, which becomes the most accessible and macroscopically damaged location of the sample.

The damage distribution at the bottom of the protected group specimens is relatively dispersed, with H_1_ and H_2_ groups and H_3_ and H_4_ groups. It is worth noting that, apart from the difference in the structure of the samples in the protected group, the damage to the H_1_ and H_2_ measuring surfaces is better than that of H_3_ and H_4_. In this paper, the brush coating method is adopted when the silane protective coating is applied. As H_1_ and H_2_ are located at the bottom of the sample, when silane is brushed, the contact between H_1_ and H_2_ at the bottom and silane is more adequate because the sample remains upright. Therefore, if conditions permit, it is recommended to use the impregnation method to protect the sample silane coating.

According to the damage zones of the sample measuring surfaces obtained in Figure 9, values were assigned to all measuring lines of the exposed group and protected group in accordance with the scoring and assignment principles in Table 7, and the total value of each measuring line after being superimposed on the four measuring lines was calculated, as shown in Table 10.

According to the assignment of each measuring line calculated in Table 10, the damaged areas of samples from the exposed group and protected group are divided according to the region division rules in Table 7, and the results are shown in Figure 10.

As can be seen from Figure 10, the exposed group samples are divided into four areas, including the mildly damaged area of three measuring line areas, including L_2_, L_3_ and L_6_; the moderate area of L_4_ and L_5_; and the main damaged area consisting of four measuring line areas, including L_7_–L_1_, in which the L_8_ measuring line area is the severely damaged area of the sample. In terms of distribution, the mildly damaged areas with relatively slight damage degree and the moderately and severely damaged areas with relatively serious damage degree show a staggered arrangement. The proportion of the mildly damaged area in the whole sample is not high, and there are areas with relatively concentrated and serious damage, indicating that the sample has been damaged to a considerable degree. The protected group samples were obviously divided into mildly damaged areas and moderately damaged areas, which accounted for roughly the same proportion in the whole sample.

From the distribution of the damaged areas, the mildly damaged area and the moderately damaged area of the protected group basically equal the whole sample, while the severely damaged area appears in the exposed group. Due to the silane coating set by the protected group in advance, its surface condition is relatively good. In the absence of original defects, it is difficult for the specimen to directly contact the corrosion products, which inhibits the generation of damage. The exposed group has the most severe and unprotected erosion environment, and its samples suffer higher damage, and moderately damaged areas appear in the exposed group. Due to the relatively strict classification conditions of the severely damaged area, at least three measuring surfaces in the position of the measuring line classified as severely damaged appeared quite serious damage. However, for samples, such as the exposed group, without erosion suppression measures, if there is a continuous moderately damaged area in the division of the damaged area, with the continuous development of the erosion process, the microcracks inside the sample will be connected somewhere in the area, thus forming a severely damaged area.

## 4. Discussion

Usually, the damage caused by the internal structure of concrete will be manifested as a decrease in ultrasonic velocity or its dynamic elastic modulus. The reason is that the damage inside the concrete is mainly caused by the expansion and development of the concrete microstructure. The expansion of the microstructure leads to the increase of the interface between the concrete and the air. When the ultrasonic wave passes through this interface, its propagation speed will decrease significantly. Then, the damage is more serious in the area where the corresponding ultrasonic velocity decreases rapidly. In addition, the images of such areas after load failure often show more crack distribution or concrete body separation. Different from mortar samples, the real damage state of the concrete samples in the process of sulfate attack will show strong non-uniformity because the performance of the aggregate in concrete will not be affected. Eliminating aggregate interference to obtain more accurately the ultrasonic velocity of mortar [32] or calculating the ultrasonic velocity of concrete by layers according to the erosion process [33] can improve the accuracy of evaluating the damage degree of concrete. However, for the real engineering structure, in addition to the overall strength decline, the uneven force caused by the difference in damage degree is also an important reason for the failure of the structure. The damaged area determined and classified according to wave velocity damage proposed in this paper can determine the difference in the structural state of concrete at the corresponding position, which is usually directly reflected in the crack propagation of concrete after loading. In order to discuss the damage delineation results and the real crack growth and development, the protected group H1 surface was taken as an example, as shown in Figure 11.

Figure 11 reflects the section features of the measuring surface before loading and the crack growth morphology after failure after loading. It is not difficult to see from Figure 11 that, thanks to the protection of silane on the concrete before loading, although the ultrasonic velocity of the specimen was damaged to some extent, there was no visible expansion crack inside the specimen, and the cracks in the section plane after failure were approximately annularly distributed around the center of the specimen. However, within the moderate damage zone of the specimen, especially in the lower sections of L_2_–L_3_ and L_6_–L_7_, the width and density of the cracks are higher, and the regional failure is more serious.

In order to analyze the results of damage delineation and the real crack growth and development, five sites were selected from the CT image results of the exposed group samples after load damage, as shown in Figure 12.

It is not difficult to see from Figure 12 that the cracks are distributed evenly in the plane with a section height of 5 cm, which is located in the middle of the sample. Under the condition of uniaxial load t, the damage degree is relatively high. The distribution of cracks in the other planes was more uneven. The cracks at the height of 1 cm were mostly concentrated on the left side and just below, the cracks at the lower left side of the section of 3 cm were the most fully developed, and the cracks at the upper right side of the section of 7 and 9 cm were seriously damaged. It is worth noting that the above areas with more serious damage belong to the moderately or severely damaged areas defined by samples from the exposed group, indicating that the sample damage evaluation and analysis method proposed in this paper is feasible to a certain extent.

## 5. Conclusions

Via ultrasonic velocity measurements and CT scanning the damage process of concrete under a sulfate erosion environment was studied. The main research results include:1.A multi-point ultrasonic test method was proposed to measure the change characteristics of sound velocity in the process of erosion deterioration. The test results showed that the samples prepared in the field showed significant difference in ultrasonic velocity change according to the material difference at the interface of measurement points;2.Based on the ultrasonic velocity damage, a concrete sulfate damage evaluation method was established, and the damage to the exposed group and the protected group samples was analyzed. The results showed that the damage to most measuring surfaces in the exposed group was more serious, and through the overall damage evaluation analysis of the samples, the damage to the protected group samples was less severe. The unevenness was weak, and the local damage to the exposed group was serious. It is shown that the silane coating set by the protected group creates a good surface condition for the specimen and inhibits damage by preventing the contact between the specimen and the corrosion products;3.Combined with the CT image analysis of concrete before and after loading, the concrete sulfate attack damage evaluation method proposed in this paper was used to divide the distribution characteristics of the damaged areas of concrete samples, and the obtained results are highly similar to the real situation.

## Figures and Tables

**Figure 1 materials-16-02658-f001:**
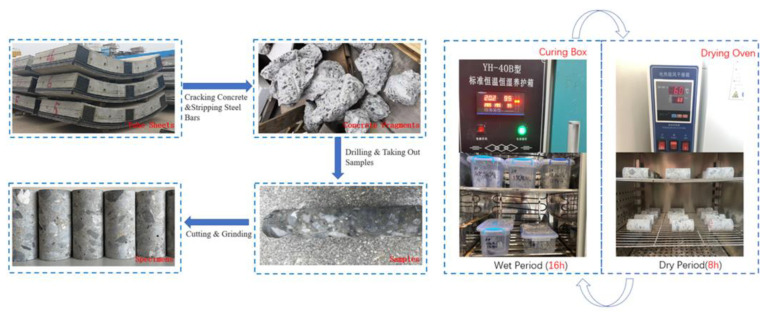
Sample preparation and test.

**Figure 2 materials-16-02658-f002:**
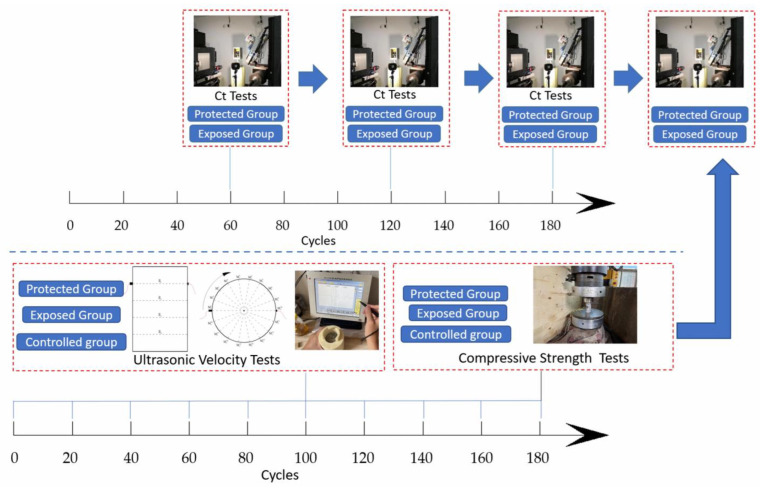
Overall test protocol of the experiment.

**Figure 3 materials-16-02658-f003:**
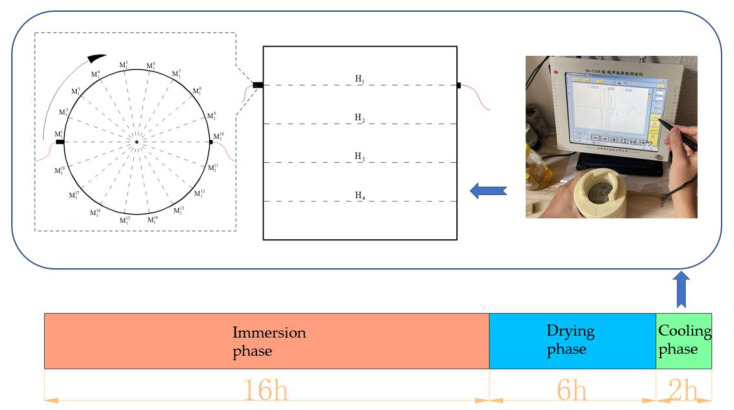
Ultrasonic velocity test scheme.

**Figure 4 materials-16-02658-f004:**
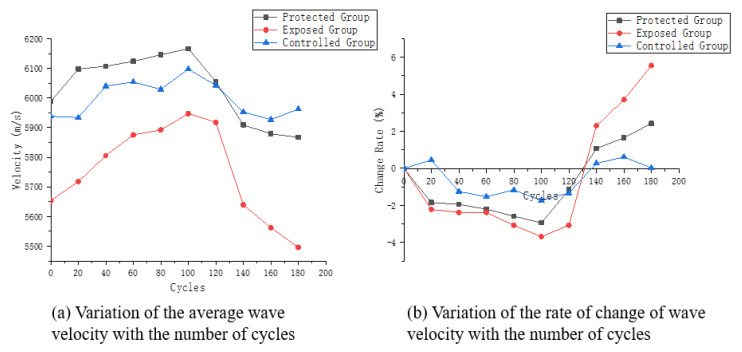
Ultrasonic velocity characteristic curve.

**Figure 5 materials-16-02658-f005:**
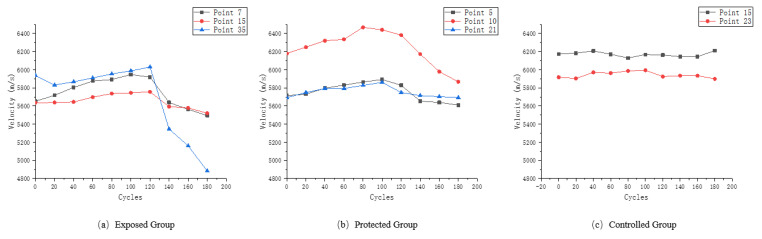
(**a**) Ultrasonic velocity variation of the characteristic measurement points of the exposed group; (**b**) ultrasonic velocity variation of the characteristic measurement points of the protected group; (**c**) ultrasonic velocity variation of the characteristic measurement points of the control group.

**Figure 6 materials-16-02658-f006:**
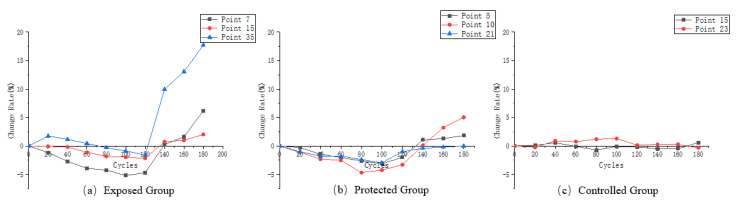
(**a**) Ultrasonic velocity variation rate of the characteristic measurement points of the exposed group; (**b**) ultrasonic velocity variation rate of the characteristic measurement points of the protected group; (**c**) ultrasonic velocity variation rate of the characteristic measurement points of the control group.

**Figure 7 materials-16-02658-f007:**
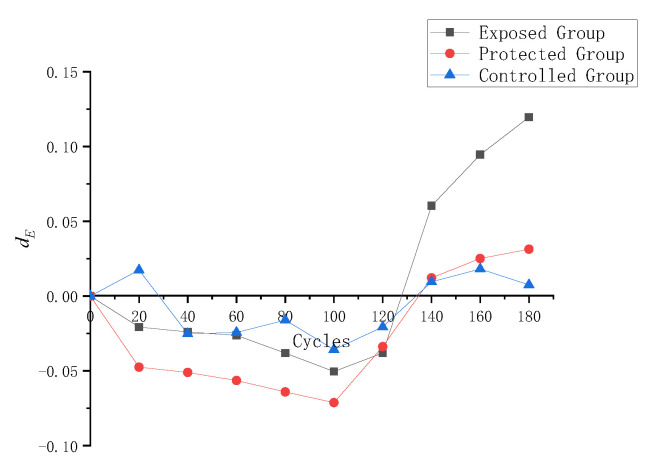
Changes in $d_{e}$ in the process of cyclic dry–wet erosion.

**Figure 8 materials-16-02658-f008:**
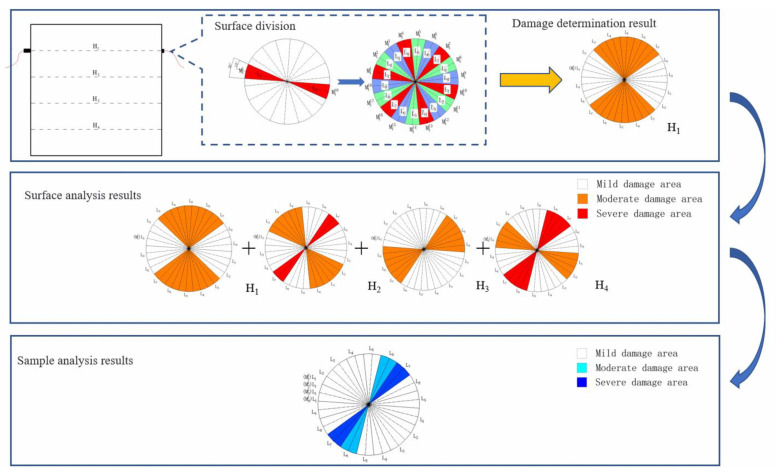
Test and analysis process. The damage region distribution of the measuring surface was determined accord-ing to the damage condition of the measuring line, and the final damage result of the sample was obtained by superimposing the damage condition of each surface.

**Figure 9 materials-16-02658-f009:**
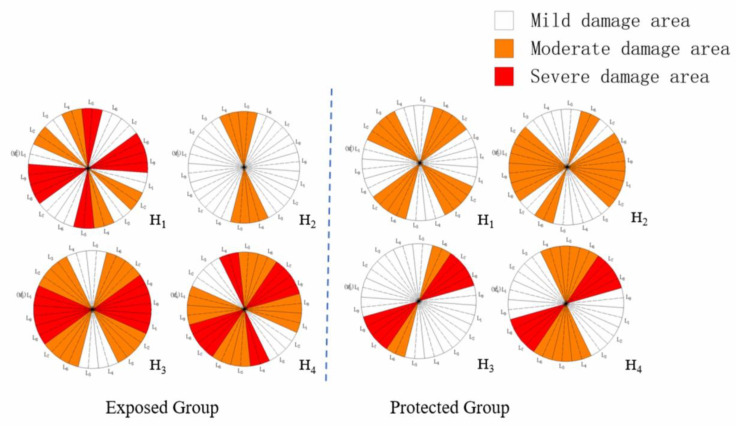
Measuring surface assessment results of damaged area.

**Figure 10 materials-16-02658-f010:**
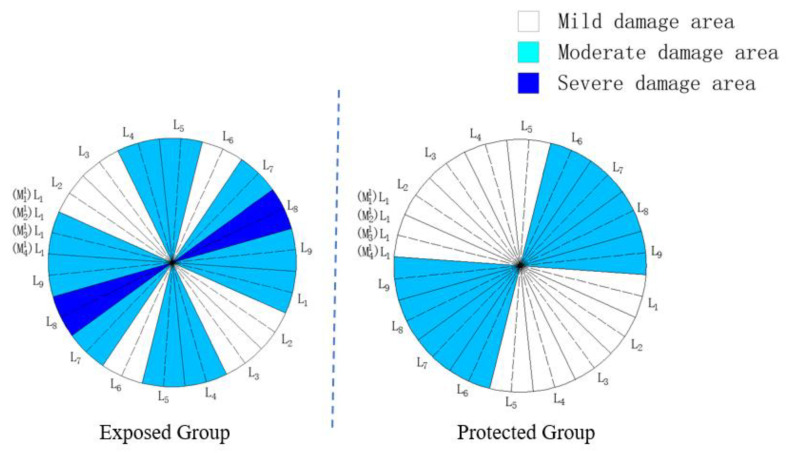
Damaged area division of exposed group and protected group.

**Figure 11 materials-16-02658-f011:**
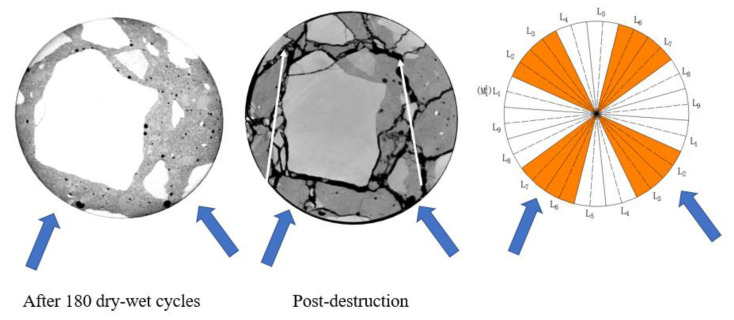
Comparison of damage classification and crack distribution of protected group H_1_ measuring surface.

**Figure 12 materials-16-02658-f012:**
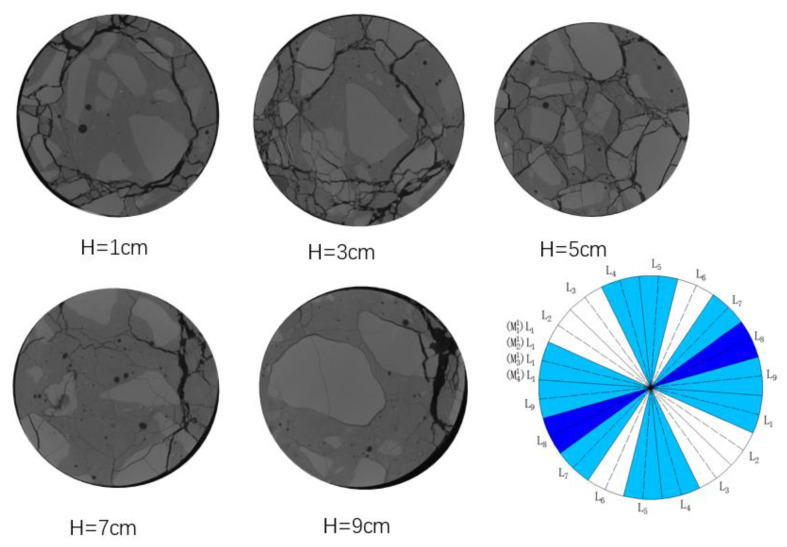
Damage patterns and damage determination results of exposed group.

**Table 1 materials-16-02658-t001:** Concrete proportion parameters (kg/m^3^).

Cement	Water	Sand	Gravel	Fly Ash	Expansion Agent	Water Reducer
410	165	711	1066	61	32.8	4.1

**Table 2 materials-16-02658-t002:** Specimen grouping and curing environment.

Group Name (Number)	Exposed Group (1)	Protected Group (1)	Control Group (1)
Environment	10% Na_2_SO_4_	10% Na_2_SO_4_, silane coating	Clear water

**Table 3 materials-16-02658-t003:** Uniaxial compressive strength of specimens.

Specimens	Exposed Group	Protected Group	Control Group
Failure load (KN)	104.58	118.79	140.42
Diameter (mm)	49.44	49.40	49.42
Uniaxial compressive strength (MPa)	54.48	61.98	73.21

**Table 4 materials-16-02658-t004:** Specimen final dynamic modulus damage.

Group	Exposed Group	Protected Group	Control Group
$d_{e}$	0.1196	0.0313	0.0075

**Table 5 materials-16-02658-t005:** Regional basis according to $R_{d}$.

Value Range	Surface Type	Damaged Area Type
$0\leq R_{d}<0.05$	Single surface	Mildly damaged area
$0.05\leq R_{d}<0.1$	Dichotomous surface	Mildly damaged area, moderately damaged area
$0.1\leq R_{d}$	Tripartite surface	Mildly damaged area, moderately damaged area, severely damaged area

**Table 6 materials-16-02658-t006:** Score of the damaged area type.

Damaged Area Type	Area Score
Mildly damaged area	0
Moderately damaged area	1
Severely damaged area	2

**Table 7 materials-16-02658-t007:** Area division according to area score.

Internal Value	Area Type
[0, 2]	Mildly damaged area
(2, 5]	Moderately damaged area
(5, 8]	Severely damaged area

**Table 8 materials-16-02658-t008:** The results of surface division.

Measuring Surface	H_1_		H_2_		H_3_		H_4_	
	$R_{d}$	Result	$R_{d}$	Result	$R_{d}$	Result	$R_{d}$	Result
Exposed group	0.1027	3	0.0864	2	0.1608	3	0.1314	3
Protected group	0.0731	2	0.0877	2	0.1458	3	0.1283	3

**Table 9 materials-16-02658-t009:** The $d_{i,j}$ value interval of the damage degree on each surface.

Group	Damage Degree	H1	H2	H3	H4
Exposed group	Mild	[0.0115, 0.0457)	[0, 0.0438)	[0.0037, 0.0573)	[0.1372, 0.1810)
	Moderate	[0.0457, 0.0799)	[0.0438, 0.0877]	[0.0573, 0.1109)	[0.1810, 0.2248)
	Severe	[0.0799, 0.1141]	/	[0.1109, 0.1645]	[0.2248, 0.2686]
Protected group	Mild	[0.0009, 0.0374)	[0, 0.0438)	[0.0247, 0.0733)	[0, 0.0428)
	Moderate	[0.0374, 0.0740]	[0.0438, 0.0877]	[0.0733, 0.1219)	[0.0428, 0.0856)
	Severe	/	/	[0.1219, 0.1705]	[0.0856, 0.1283]

**Table 10 materials-16-02658-t010:** Assignment statistics on each measuring line.

Group		L_1_	L_2_	L_3_	L_4_	L_5_	L_6_	L_7_	L_8_	L_9_
Exposed group	H_1_	0	1	0	1	2	0	0	2	2
	H_2_	0	0	0	1	1	0	0	0	0
	H_3_	2	1	1	0	0	1	1	2	2
	H_4_	1	0	0	2	1	1	2	2	1
	Sum	3	2	1	4	4	2	3	6	5
Protected group	H_1_	0	1	1	0	0	1	1	0	0
	H_2_	1	1	0	0	0	1	0	1	1
	H_3_	0	0	0	0	0	1	2	2	0
	H_4_	0	0	0	0	1	1	1	2	2
	Sum	1	2	1	0	1	4	4	5	3

## Data Availability

The data that support the findings of this study are available upon request from the authors.

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
