# Peer review of "Compartmentalized Quantitative Analysis of Concrete Sulfate-Damaged Area Based on Ultrasonic Velocity"

_materials, 2023, doi:10.3390/ma16072658_

Round 1

Reviewer 1 Report

Yinghua Jian has reported manuscript with tile Compartmentalized quantitative analysis of concrete sulfate 2 damage area based on ultrasonic velocity.The paper is well written and have good application. I recommended it after changes.

1. Revised abstract and introduction. 

2. Why author use ultrasound technology?

3. how ultra sound technology practical applicable?

4. There is missing statistical formulas on data 

5. There is need to update references. 

Author Response

Thank you for your constructive remarks and useful comments. We have carefully considered the comments and made some modifications.

Reviewer 2 Report

Authors have proposed a compartmentalized quantitative analysis of sulfate damage in concrete using ultrasonic velocity validated by CT scans. The study is interesting and deserves publishing in this journal. The following comments could further improve the quality of the manuscript.

1. English language is good, however, another round of revision is needed to enhance the readability of the manuscript. 

2. Caption of the Figures is very short. For instance, the caption of Fig. 2 needs more explanation.

3. After Figure 2 (line 137), there is a caption for Table 4, but Table 4 is missing! please check and fix that.

4. The quality of plots in Figure 4 is very low and should be improved. Use a quality similar to Fig. 7.

5. Line 272, FIG.6 should be changed to Fig. 6. Please check this for the entire manuscript.

6. In line 355, should it be Table 4 or Table 5?

7. In Table 8, Rd and Result should be moved to the subheadings of the Table (under H1, H2,...).

8. Abstract and Conclusions need to be revised to better reflect the finding of this study. 

Author Response

We havThank you for your constructive remarks and useful comments. We have carefully considered the comments and made some modifications.e tried our best to revise the paper with considering your comments.

Reviewer 3 Report

Paper ID: materials-2233580

Type:Article
Title: 
Compartmentalized quantitative analysis of concrete sulfate damage area based on ultrasonic velocity

Authors: Yinghua Jian , Dunwen Liu , Kunpeng Cao * , Yu Tang

 This paper investigates compartmentalized quantitative analysis of concrete sulfate damage area based on ultrasonic velocity. Although the testing methods and compared results attained in the present study show the importance of the paper, the authors should address the following comments:

1.     Novelty in comparison to recent literature? Need to be emphasized in the last paragraph of Introduction section.

2.     There should be space between the number and the unit.

3.     Please revise subscripts and superscripts throughout the manuscript.

4.     Why did the authors choose C50?

5.     The chemical and physical properties of cement and fly ash might be given, as well as particle size distribution.

6.     The relevant literature must discuss the results in the paper.

  1. Throughout the text, some typos must be eliminated.
  2. Please give the standard deviation of the results.

Author Response

(The authors gave the same response as above.)

Round 2

Reviewer 3 Report

The authors have made the necessary changes. Therefore the manuscript can be accepted.

Author Response

According to the review opinions, we went through the full text again and modified some expressions of the manuscript to improve the readability of the article. At the same time, we also corrected some mistakes in the manuscript.